# Oral PRI-002 treatment in patients with MCI or mild AD: a randomized, double-blind phase 1b trial

Self-replicating amyloid beta (Aβ) oligomers are considered as one of the major drivers for disrupted synaptic function and plasticity, leading to impaired neuronal viability and progression of Alzheimer's disease (AD). Here, we investigated the safety, tolerability and pharmacokinetics of the anti-oligomeric peptide PRI-002, which was developed to disassemble toxic Aβ oligomers into non-toxic monomers. In a randomized, double-blind, single-center phase 1b trial, 20 patients aged between 50 and 80 years, with mild neurocognitive impairment (MCI) or mild dementia due to AD were recruited. Eligible patients were randomly assigned (1:1) to receive 300 mg PRI-002 once daily (q.d.) or placebo for 28 days. During treatment, study visits were performed on baseline (Day 1), Day 14, Day 28 and an additional follow-up visit on Day 56. Safety assessments were carried out at all visits to determine the primary endpoints. On Day 7 and Day 21 additional phone visits were carried out to assess concomitant meds and AEs. Primary endpoints were nature, frequency, severity, and timing of adverse and serious adverse events (AE/SAEs) and treatment discontinuation. Furthermore, standard laboratory values, electrocardiogram (ECG), electroencephalogram (EEG), magnetic resonance imaging (MRI), and vital signs were assessed. Secondary endpoints included the evaluation of pharmacokinetic characteristics of PRI-002 in plasma and the determination of cerebrospinal fluid (CSF) concentrations of PRI-002. The trial is registered in EudraCT 2020-003416-27 and clinical-trials.gov NCT04711486. In the study, 19 out of 20 patients were randomly assigned to PRI-002 (n = 9) or placebo (n = 10) and completed the study. One patient withdrew informed consent before randomization. All primary endpoints were met. Overall, the study drug was well tolerated. In total n = 16 AEs were reported in the verum group, while n = 27 AEs were noted in the placebo group. No SAEs were reported. No significant changes in clinical chemistry, hematology or hematoserology were detected. ECG, EEG and MRI revealed no changes and in detail no ARIA were observed. Pharmacokinetic parameters were unrelated to sex, age, and weight. Furthermore, no significant changes were detected in p-tau, t-tau, Aβ 1-40, Aβ 1-42 and Aβ oligomers in CSF. Patients receiving PRI-002 performed significantly better than those receiving placebo

✉ e-mail: d.willbold@fz-juelich.de; oliver.peters@charite.de

in the CERAD word list at Day 56 (P ≤ 0.05). In conclusion, 28 days of treatment with 300 mg q.d. PRI-002 was well tolerated in patients with MCI or mild dementia due to AD.

After two decades of setbacks, the United States Food and Drug Administration (USFDA) full approvals for lecanemab[1] and donanemab[2] presents the beginning of a new decade for AD patients with several more drug candidates holding promise for marketing authorisation in the foreseeable future[3]. Although the disease is still far from being cured, scientific evidence suggests the elimination of aggregated Aβ will at least attenuate the progression of clinical symptoms in early AD patients[4]. Currently, only anti-amyloid therapeutic antibodies (aducanumab[5,6], lecanemab[7], and donanemab[2]) have received conditional or full approval, validating Aβ aggregates as one of the most important targets and intervention points for the treatment of AD. Controversy is left on what species of which Aβ aggregates are most beneficial to be targeted. The clinical benefit of aducanumab, lecanemab, and donanemab, are up to 36% deceleration of progression measured by CDR-SB versus placebo, over 18 months, on average[2,4,7–10]. This clinical benefit comes with an impressive plaque load reduction as monitored by amyloid PET, but also with side effects like amyloid-related imaging abnormalities (ARIA).

Aside from amyloid plaques as a characteristic pathological feature of AD, self-replicating Aβ oligomers are described to be synaptotoxic and responsible for reduced synaptic function and plasticity, impaired neuronal function and thus for development and progression of AD. The all-D-enantiomeric peptide PRI-002, also named "RD2" or "Contraloid" in previous publications and regulatory documents, was developed to disassemble toxic Aβ oligomers into harmless Aβ monomers[11]. Thus, PRI-002 can be expected to reduce neurotoxicity and to restore synaptic plasticity in early AD stages. Target engagement has been demonstrated in vitro[12], in vivo[13] and ex vivo[11]. PRI-002 has demonstrated safety and tolerability in healthy volunteers with suitable pharmacokinetic characteristics to support further clinical development[14]. Earlier, we demonstrated that PRI-002 is robustly improving memory and cognition in animal models[12,13,15,16] in contrast to only decelerate decline. A possible explanation for this observation is that PRI-002 is indeed reducing the synaptotoxic effects of soluble Aβ aggregates (oligomers) directly at the synapse and the neuron by disassembling them into non-toxic monomers. This would plausibly explain that synapses and neurons can become functional again, and thus allow improvement of neuronal function shortly after initiation of treatment. This observation in animal models needs to be validated in human patients. Disease modification on the long run and immediate functional improvement may sum up to high efficacy which cannot be achieved to the same extent by anti-amyloid antibodies.

Here, we evaluate the safety, tolerability and pharmacokinetics in patients with MCI or mild dementia due to AD in a single-center, randomized, placebo-controlled, double-blind, phase 1b study. After 28 days of treatment PRI-002 was safe in patients with early stages of the AD continuum. While no neurochemical biomarker changes were detected during the study, all 9 verum treated patients showed an improvement of memory function in the CERAD word list test performance at Day 56 assessment 4 weeks after the end of treatment.

## Results

### Study design and participants

Between December 8, 2020, and January 13, 2022, 23 patients were screened and assessed for eligibility. 19 patients were randomized and allocated to trial.

Patients received once daily oral doses of 300 mg PRI-002 or placebo for 28 days. Safety and efficacy assessments were performed at baseline, Day 14, Day 28 during the treatment period and Day 56 at the last visit. 10 patients (age 76.9 ± 3.4, MMSE 28 ± 1.6) received placebo and 9 patients received PRI-002 (age 72.4 ± 6.9, MMSE 27.2 ± 2.9). There were no significant differences in the patient's baseline characteristics between the placebo and PRI-002 group (Table 1).

### Safety

13 patients reported AEs. The overall incidence of AEs regardless of its relationship to study treatment and the incidence of AEs considered probably or possibly related to study treatment are presented in Table 2.

Overall, in the safety population (n = 9 PRI-002, n = 10 placebo), 56% (5/9) of subjects in the PRI-002 group and 80% (8/10) of subjects in the placebo group had one or more AEs (Table 2). The severity of the AEs has been reported for 40 AEs as mild and for three as moderate with mild (grade 1), 14 PRI-002 versus (vs.) 26 placebo and moderate (grade 2), two PRI-002 vs. one placebo. Five patients of the PRI-002 treatment reported 16 AEs and eight patients of the placebo group had 27 AEs. The most commonly reported AEs were nervous system disorders (seven in PRI-002 group vs. one in placebo group); psychiatric disorders (one in PRI-002 group vs. five in placebo group) and vascular disorders (one in PRI-002 group vs five in placebo group). More precisely one patient in the verum group reported six times short-term dizziness on Day 4, Day 8, Day 9, Day 10, Day 18 and Day 20. Another patient in the verum group reported once slight fatigue combined with short-term dizziness on Day 13 and twice fatigue simultaneous to administration of Donepezil on Day 20 and Day 21. Altogether the described events were rated as probably related but not considered as critical in any regard. Especially there is no evidence of an increase of number or severity of adverse events

**Table 1 | Demographic data, cognitive measures and biomarker level at baseline**

|  | Placebo (n = 10) | PRI-002 (n = 9) |
|---|---|---|
| **Demographic data** | | |
| Female sex, n (%) | 6 (60%) | 5 (55.6%) |
| Age, years (SD) | 76.9 (3.5) | 72.4 (7.0) |
| **APOEε4 carriers** | 1 (10%) | 2 (22.2%) |
| **Cognitive measures** | | |
| MMSE | 28 (1.6) | 27.2 (3.0) |
| WL learn | 14.3 (3.7) | 15.4 (4.6) |
| CDR-SB | 1.56 (1.01) | 2.93 (2.26) |
| **CSF biomarker** | | |
| p-tau$_{181}$ [pg/ml] | 85.3 (30.2) | 110.9 (59.7) |
| t-tau [pg/ml] | 543.5 (159.5) | 685.2 (290.6) |
| Ratio Aβ 42/40 | 0.044 (0.011) | 0.046 (0.010) |
| Aβ 1-42 oligomers [fM] | 5.059 (11.474) | 4.088 (6.096) |

Sex, age, APOEε4 and MMSE status were determined at the initial screening visit (Day 0), cognitive measures and CSF-values were obtained at the baseline visit (Day 1). Data are n (%), mean (SD). Source data are provided as a Source Data file.
*MMSE* Mini-Mental State Examination, *WL learn* CERAD (Consortium to Establish a Registry for Alzheimer's Disease) Word list learning, *CDR-SB* Clinical Dementia Rating Sum of Boxes, *Aβ* amyloid beta.

over time. A detailed listing of the clinical AEs can be found in supplement (see Supplementary Table 2). No SAEs or deaths were reported during the trial.

In addition, there were no clinically relevant changes detected in laboratory parameters including clinical chemistry, blood count or hematoserology from screening to midterm visit to end of treatment in both groups (all parameters determined are listed in Supplementary Table 3 and Supplementary Table 9). There were also no significant

changes on vital signs, in general physical examination, in ECG and EEG (Supplementary Tables 10–12) assessment or in any of the tested biomarkers (p-tau, t-tau, Aβ 1-42, Aβ 1-40, and Aβ oligomers) (Table 3). Also, MRI did not reveal safety relevant changes (Supplementary Table 13). In contrast to the ARIA like microhaemorrhages or vasogenic edema reported after treatment with anti-β-amyloid monoclonal antibodies[17], no edematous changes occurred after PRI-002 treatment. One new isolated microbleed and an approx. 2 mm large bifurcation aneurysm were detected in two placebo patients.

## Pharmacokinetics

Supplementary Table 4 shows the statistics of PRI-002 pharmacokinetic parameters in MCI- and Alzheimer's patients. On the two test days (Day 1 and Day 28), plasma levels were highly variable (Fig. 1). Coefficient of variations (CVs) were clearly above 100% at 0.5 h after the first treatment and at the pre-treatment time of Day 28 (Supplementary Table 8). Other samples showed CVs of close to 100% and a CV of below 60% was not observed (Supplementary Table 8). The high variability is due to a non-Gaussian data distribution. Mean maximum plasma levels on Day 1 (Day 28) of $4.46 \pm 3.71$ ng/ml ($12.6 \pm 13.2$ ng/ml) were reached after about 2 h (1 h) and mean $AUC_{0-4\ hours}$ were $10 \pm 7.5$ ng*h/ml ($26.8 \pm 23.8$ ng*h/ml). Generally, pharmacokinetic parameters showed high variability characterized by coefficients of variation ranging from 77% to 134%. Sampling did not allow for the calculation of $t_{1/2}$. Therefore, the percentage of the partial AUC up to 4 h as calculated for Day 1 of the MAD[14] study ($AUC_{0-4\ hour} = 33.7$ ng*h/

## Table 2 | Summary of adverse events

| | Placebo (n = 10) | PRI-002 (n = 9) | Total (n = 19) |
|---|---|---|---|
| **Overall incidence of Adverse Events (AEs)** | | | |
| Number of subjects with at least one AE (%) | 8 (80%) | 5 (56%) | 13 (68%) |
| Mild (grade 1) | 26 | 14 | 40 |
| Moderate (grade 2) | 1 | 2 | 3 |
| Severe (grade 3) | 0 | 0 | 0 |
| **AEs probably or possibly related to study treatment** | | | |
| Number of subjects with at least one treatment-related AE (%) | 1 (10%) | 1 (11%) | 2 (10.5%) |
| Mild (grade 1) | 1 | 4 | 5 |
| Moderate (grade 2) | 0 | 2 | 2 |
| Severe (grade 3) | 0 | 0 | 0 |

## Table 3 | Statistics for Biomarkers

| Biomarker | Timepoint | Placebo | | | PRI-002 | | | PV | PV-LME |
|---|---|---|---|---|---|---|---|---|---|
| | | Mean | SD | n | Mean | SD | n | | |
| p-tau [pg/ml] | Day 1 | 85.3 | 30.2 | 10 | 110.9 | 59.7 | 9 | 0.50 | 0.785 |
| | Day 28 | 88.5 | 30.4 | 10 | 112.4 | 60.8 | 9 | 0.50 | |
| t-tau [pg/ml] | Day 1 | 543.5 | 159.5 | 10 | 685.2 | 290.6 | 9 | 0.32 | 0.846 |
| | Day 28 | 565.9 | 154.1 | 10 | 698.7 | 305.0 | 9 | 0.32 | |
| Aβ42 [pg/ml] | Day 1 | 517.5 | 180.3 | 10 | 575.8 | 167.1 | 9 | 0.60 | 0.858 |
| | Day 28 | 523.0 | 184.8 | 10 | 573.9 | 205.8 | 9 | 0.50 | |
| Ratio Aβ42/40 | Day 1 | 0.044 | 0.011 | 10 | 0.046 | 0.010 | 9 | 0.84 | 0.910 |
| | Day 28 | 0.042 | 0.009 | 10 | 0.044 | 0.011 | 9 | 0.90 | |
| Aβ42 oligomers [fM] | Day 1 | 5.059 | 11.474 | 10 | 4.088 | 6.096 | 9 | 0.720 | 0.589 |
| | Day 28 | 6.377 | 14.948 | 10 | 4.747 | 6.811 | 9 | 0.462 | |

Table with statistics for CSF biomarker. P-values for one time point (column 'PV') were calculated using two-sided Wilcoxon test. PV for the comparison of longitudinal effects between both groups (Placebo vs PRI-002) (column 'PV-LME') where calculated with mixed linear models. Day 1: baseline visit; Day 28: End of Treatment visit. Source data are provided as a Source Data file.

*Aβ* Amyloid beta; *n* number, *p-tau* phospho tau, *PV* P value, *SD* standard deviation, *t-tau* total tau.

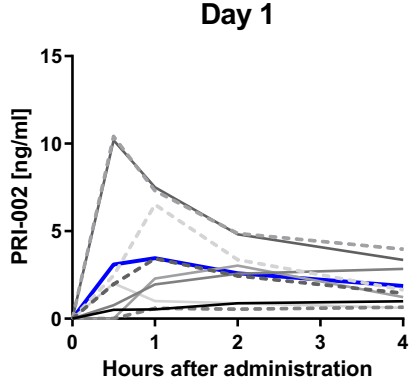
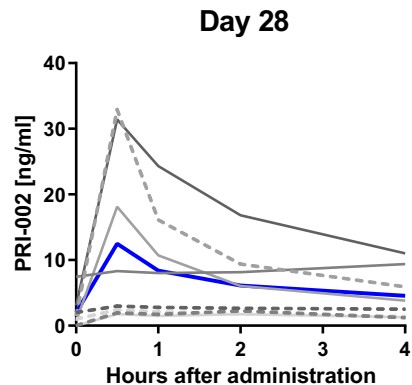

**Fig. 1 | Individual and mean plasma concentrations of PRI-002 over time on day 1 and day 28.** Blue line: mean plasma concentration of PRI-002 (*n* = 9); gray lines: individual plasma concentrations Source data are provided as a Source Data file.

ml corresponding to 44.2% of total AUC; dose: 320 mg/person) was used to extrapolate the total AUC for the present patient study: 22 ng*h/ml. Using the quotient $AUC_{0-4\ hour}$ Day 28/$AUC_{0-4\ hour}$ Day 1 the mean systemic drug increase was $2.5 \pm 1.3$ fold. No correlations were found between any pharmacokinetic parameter and sex, age, or weight of patients.

In addition, also CSF levels of PRI-002 were determined. Due to non-specific binding of PRI-002 to vials used for sampling and storage the measured values, which are depicted in the Supplementary Table 7 are minimal values and the actual values will most probably be higher than reported here. Considering the three CSF samples in which the PRI-002 levels were above the LLOQ, approximately 8% of the mean predose plasma level of patients 2, 3 and 4 at Day 28 ($3.74 \pm 3.42$ ng/ml) reached the CSF.

### Exploratory cognitive assessment

The CERAD+ battery is a widely used cognitive test battery for the evaluation of multiple cognitive domains such as episodic memory, language, and visuospatial ability and a valid measure of cognitive progression in AD[18]. In this study the analysis of the different tests of the CERAD+ battery (for all data see Supplementary Table 5) revealed a significant change in the word list learning test (Table 4). The Cohen's d effect size for day 56 was ES:0.94, indicating a large effect and therefore supporting a true treatment effect. The word list, within the CERAD+ battery may be used for changes in short term memory capabilities also within the short time frame of 56 days. The analysis of the longitudinal progression of the CERAD values revealed a significant improvement of the PRI-002 treated patient between baseline and follow up (day 56) and end of treatment (day 28) and follow up (day 56), which was not observed in the placebo treated patients (Fig. 2B). At day 56 (follow up) also the group comparison became significant (Fig. 2A). Figure 2C shows that the performance of every PRI-002-treated patients improved, which is in contrast to the placebo-treated patients, some of whom improved, some of whom stayed the same and some of whom worsened. In addition, we analyzed a possible relationship between the number of patients treated with cholinesterase inhibitor and the duration of treatment on one hand and the improvement of memory function as measured by the CERAD word list (Supplementary Table 6), on the other hand.

Based on the heterogeneous individual oral bioavailabilities among patients receiving PRI-002, we analyzed the data to determine possible pseudo-dose-response relationships. A significant inverse relationship was observed between changes in Aβ oligomer concentrations (given as slopes from Day1 to Day 28) and the PRI-002 plasma levels at Day 1 (Fig. 3).

## Discussion

The phase 1b study reported here demonstrated safety, tolerability and an improvement of short-term memory function in the CERAD word list learning of PRI-002, which was developed for the disease-modifying therapy of AD by disassembling toxic Aβ oligomers. In recent years, Aβ oligomers have been validated multiple times as a key target for AD therapy, as monoclonal antibodies selectively targeting aggregated forms of Aβ, including soluble oligomers and insoluble fibrils, are the first disease-modifying therapies for AD that have been shown to slow clinical decline by interfering with fundamental biological processes of the disease[19]. Furthermore, it has become clear that the specificity respectively the exact binding preference for an Aβ species determines both efficacy and risk for potentially serious adverse events like ARIA-E or ARIA-H[20], as lecanemab, for which it has been claimed to preferentially target soluble Aβ protofibrils, has the lower risk-benefit-ratio compared to aducanumab, which targets Aβ oligomers and Aβ plaques[21]. Lecanemab treatment reduced amyloid burden in early AD and resulted in a slowed decline by 27% on measures of cognition and function (CDR-SB score, ADAS-cog14 score, ADCOMS and ADCS-MCI-ADL score) than placebo at 18 months[4].

In order to more convincingly place the results of this study with PRI-002 in the context of the antibody treatment studies, the results of the antibody studies after shorter treatment durations must be considered. Interestingly lecanemab 10 mg biweekly has also shown a symptomatic improvement as measured by ADAS-cog from week 12[22] and most prominent at 6 months after treatment was initiated[7]. Donanemab (700 mg for the first 3 doses and 1400 mg thereafter administered intravenously every 4 weeks) even demonstrated a significant improvement after 12 weeks of treatment in the ADAS-Cog[13] and the CDR-SB. These clinical efficacy signals, demonstrated by improvement in symptoms combined with significant changes in biomarkers, provide the first evidence of a symptomatic effect that results from disease modification[22]. The collective data suggest that short-term effects can be achieved by targeting synaptotoxic forms of Aβ.

Further development in antibodies against Aβ species is ongoing interestingly focusing on Aβ oligomers, which strengthens our view to put these kinds of amyloid species into focus[23].

**Table 4 | Statistics for functional outcome measures**

| Variable | Visit | Placebo | | | PRI-002 | | | PV | PV-LME |
|---|---|---|---|---|---|---|---|---|---|
| | | Mean | SD | n | Mean | SD | n | | |
| WL learn | Day 1 | 14.30 | 4 | 10 | 15.44 | 5 | 9 | 0.743 | 0.036 |
| | Day 28 | 15.33 | 4 | 9 | 17.25 | 5 | 8 | 0.923 | |
| | Day 56 | 15.50 | 3 | 10 | 19.33 | 5 | 9 | 0.036 | |
| WL Recall | Day 1 | 2.4 | 1.7 | 10 | 4.7 | 2.3 | 9 | 0.04 | |
| | Day 28 | 3.2 | 2.2 | 9 | 6.0 | 3.3 | 8 | 0.07 | 0.17 |
| | Day 56 | 4.2 | 1.4 | 10 | 5.4 | 2.2 | 9 | 0.39 | |
| CDR-SB | Day 1 | 1.56 | 1.01 | 9 | 2.93 | 2.26 | 7 | 0.26 | |
| | Day 28 | 1.31 | 0.46 | 8 | 1.75 | 1.89 | 8 | 0.75 | 0.484 |
| | Day 56 | 1.79 | 0.70 | 7 | 2.39 | 2.23 | 9 | 0.96 | |
| MMSE | Day 1 | 26.9 | 1.91 | 10 | 27.1 | 2.93 | 9 | 0.535 | |
| | Day 28 | 26.4 | 2.40 | 9 | 27.9 | 2.36 | 8 | 0.261 | 0.730 |
| | Day 56 | 27.3 | 1.83 | 10 | 27.2 | 2.39 | 9 | 0.900 | |

*P* values (PV) for one time point (column 'PV') were calculated using two-sided Wilcoxon test. PV for the comparison of longitudinal effects between both groups (Placebo vs PRI-002) (column 'PV-LME') where calculated with mixed linear models. Day 1 is the baseline visit, Day 28 is the end of treatment visit and Day 56 is the End of Study visit. Source data are provided as a Source Data file.
*n* number, *SD* standard deviation, *WL learn* CERAD Word list learning, *WL Recall* Word list recall test, *CDR-SB* Clinical Dementia Rating scale Sum of Boxes l, *MMSE* mini mental state examination.

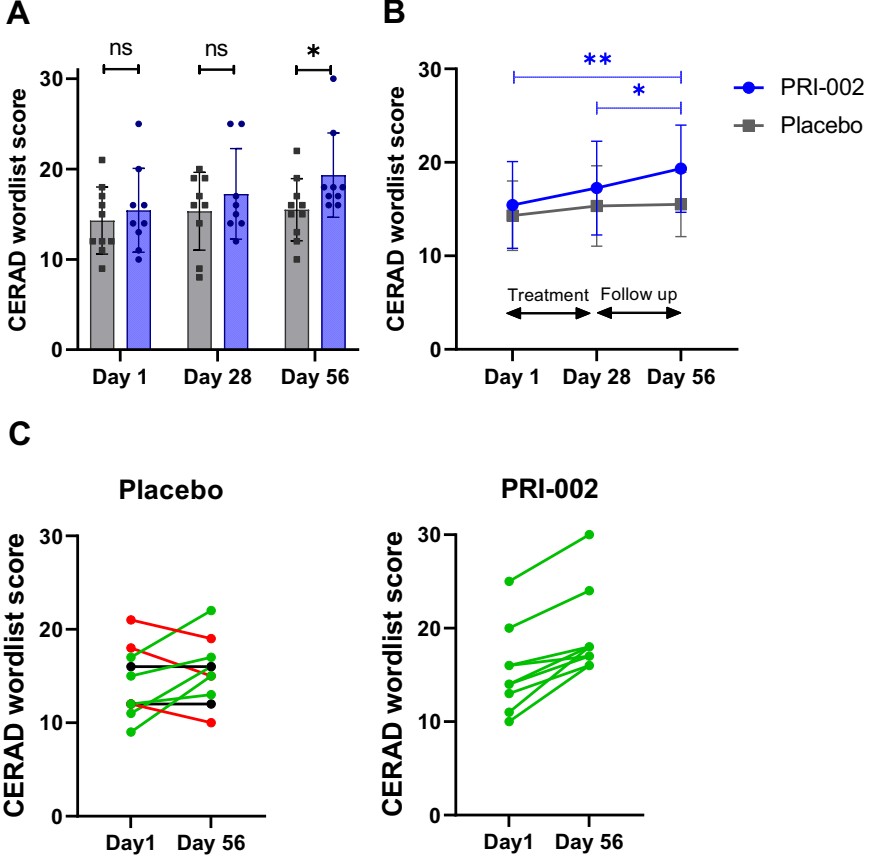

**Fig. 2 | CERAD word list learning. A** Bar graph shows the mean values of the CERAD word list scores with SD and individual data points of the individual patients. (gray: placebo, $n = 10$ individual patients; blue: PRI-002, $n = 9$ individual patients). *P* values (two-tailed) for different time points were calculated using two-sided Wilcoxon test (*$p = 0.036$; ns: not significant). **B** Line graph with mean values of the CERAD word list scores and SD of the CERAD word list values over time (blue: PRI-002 $n = 9$ individual patients, gray: placebo $n = 10$ individual patients). A two-way repeated measures (RM) ANOVA was performed $F_{timepoint, treatment}$ $(2,17) = 2.364$ $p = 0,110$, $F_{timepoint}$ $(2,17) = 8.093$, $p = 0,001$; $F_{treatment}$ $(1,17) = 1.586$,

$p = 0,225$. Subsequent a post-hoc Holm-Sidak test was performed **$p < 0.001$, *$p = 0.022$. **C** Line plot of the CERAD word list learning values (one value for each patient per timepoint) over time for each patient in the two groups between Day 1 (baseline prior the first administration of the study medication) and Day 56 (end of study). *P* values were calculated using two-sided Wilcoxon test. p-values placebo: 0.36; PRI-002: 0.009, color code: green = increase of performance; black = no change; red = decrease of performance for individual patients. (placebo, $n = 10$ individual patients; PRI-002, $n = 9$ individual patients). Source data are provided as a Source Data file.

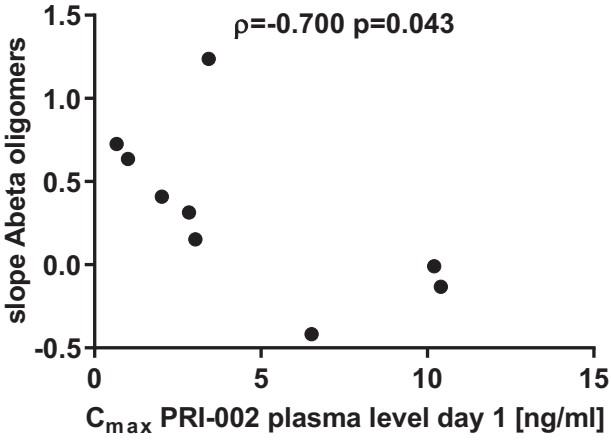

**Fig. 3 | Scatterplots and correlation coefficients including p-values for slope Aβ oligomers and PRI-002 plasma level on Day 1.** The slope corresponds to a difference normalized by visit (0.5 * Δ Day56 – Day 1). Correlations were performed with Spearman (ρ) analysis (two-tailed) at alpha level 0.05 with $n = 9$. Source data are provided as a Source Data file.

Synapse loss correlates with cognitive impairment[24] and is also evident during early stages of AD[25], however numerous studies have failed to confirm a correlation between the amount of amyloid plaques burden and the severity of dementia or the loss of neurons and synapses[26–28]. Therefore, it can be postulated, that treatment must halt synapse loss in order to improve cognitive impairment in the early stages of the disease. PRI-002 was developed to disassemble toxic Aβ oligomer into non-toxic Aβ monomers and thereby restoring synaptic function and plasticity. With this mode of action, positive effects on cognition are already feasible in the early stages of the disease.

Here, we report results from the randomized, double-blind, placebo-controlled ContraloidAD trial that assessed primarily safety, tolerability and pharmacokinetics of the all-D-enantiomeric peptide PRI-002. This study was the first clinical trial collecting safety parameters of PRI-002 in patients with MCI and mild dementia due to AD, including data on neurochemical biomarkers like Aβ40, Aβ42, tau, ptau, and Aβ oligomer concentrations in CSF as well as cognitive measures. The very limited sample size ($n = 19$) in combination with the very short treatment duration of 28 days, however, narrows the scope of the analysis of these secondary outcomes to hypothesis generation for the outcome of future efficacy trials.

This phase 1b study met its primary prespecified outcomes with regard to safety and tolerability in patients in early stages of the AD

continuum. Since PRI-002 belongs to a relatively new class of drugs, the all D-enantiomeric peptide compounds, which combine the advantages of small molecules (high protease resistance, low immunogenicity, chemically synthesizable, blood brain barrier-permeability) with those of biologicals (high target specificity and affinity), these results are in line with expectations.

No significant changes in any measured biomarker was observed between Day 28 versus baseline or verum versus placebo at Day 28. The very small individual changes of Aβ oligomer concentrations within the verum group, however, correlate strong and significantly with the individual PRI-002 levels in blood at baseline (Fig. 3). The observation that only three out of nine verum patients showed an absolute reduction of Aβ oligomer concentration in CSF (negative slope) can potentially be interpreted towards the conclusion that the dose and/or the treatment duration were too low. This will be investigated in the ongoing phase 2 study (https://clinicaltrials.gov/study/NCT06182085) with higher dose and longer treatment duration. For the interpretation of any biomarker changes from baseline to day 28, one has to keep in mind that biomarkers usually do not change within 4 weeks, but within months and years, and that biomarkers measured from blood and also from CSF are peripheral to the brain and this need time to reflect changes in the brain. The mean $C_{max}$-value at day 28 ($12.6 \pm 13.2$ ng/ml) was in the concentration range that resulted in significant improvement in cognitive performance of 200 mg/kg treated old aged APP/PS1delta E9 mice (range of $1.5$ ng/ml $\pm 7.5$ ng/ml)[29]. In previous studies of PRI-002 (alias RD2) in AD animal models[12,13,15,16], improvement of memory and cognition was demonstrated as a very robust treatment outcome. The word list score at Day 56, that is 4 weeks after treatment had ended, a significant improvement of learning and short-term memory function was observed in the CERAD word list learning in the verum group compared to baseline and compared to the placebo group. The Cohen's d effect size for day 56 was ES:0.94, indicating a large effect and therefore supporting a true treatment effect. One may speculate, whether PRI-002 mediated disassembly of Aβ oligomers in monomers, as demonstrated ex vivo[11], was beneficial for the functionality of synapses and neurons in the verum patients, before the concentration of Aβ oligomers in relatively peripheral CSF had a chance to become significantly reduced, too. Irrespective, whether this reflects a disease modification, it is quite promising that no immediate deterioration of cognitive function was observed upon PRI-002 treatment, like it was observed with other treatment approaches i.e. with secretase inhibitors[30]. We propose that PRI-002 with its specific mode of action targeting synaptotoxic oligomers, has the potential to rescue synapses in early disease stages, and by this restores memory function of patients with mild symptoms of AD. Longer treatment with more patients is needed to investigate this. A phase 2 study has started in 2024 (https://clinicaltrials.gov/study/NCT06182085).

Disease modifying small molecules like PRI-002 may outweigh treatment with biologics like anti-Aβ-antibodies because they are orally available, do not trigger auto immune responses, do not cause ARIAs, and therefore have less site effects and might be suitable for the long-term or even preventive treatment.

## Methods
### Study design
This study was conceptualized as a single center, randomized, double-blind, placebo-controlled trial with a parallel-group design. Participants had to fulfill the clinical criteria for MCI due to AD, according to DSM-V or mild dementia due to AD according to ICD-10. Patients were recruited at the Memory Clinic of the Charité Universtitätsmedizin Berlin (Germany), Department of Psychiatry and Neuroscience and had experienced a routine assessment at the memory clinic and a pre-screening procedure prior to screening for the study. Implementation of the prescreening assured screening failure rates as low as possible.

The study protocol and consent forms were approved by the local authorities (Landesamt für Gesundheit und Soziales, LaGeSo). All participants were fully capable to provide and have signed a written informed consent. The study was performed in accordance with the Declaration of Helsinki and the principles of Good Clinical Practice as described in the International Council for Harmonization guidelines. The trial is registered in EudraCT 2020-003416-27 and clinicaltrials.gov NCT04711486.

### Participants
Twenty male and female (not of childbearing potential) patients between 50 and 80 years of age, with a Mini mental state examination (MMSE) score of 22 to 30, CSF biomarkers indicating AD pathology (p-tau >62 pg/ml, total CSF Aβ 1-42/1-40 ratio ≤0.055), a MRI scan in accordance with AD diagnosis not older than 3 months, at least 3 months stable medication prior to screening, were enrolled. Nineteen patients were randomized and data analyzed, one patient dropped out by withdrawal of the informed consent before treatment was initiated. Patients underwent an extensive clinical workup and received a MRI brain scan as well as a lumbar puncture. A neuropsychological examination using the CERAD+ (Consortium to Establish a Registry for Alzheimer's Disease plus) test battery was performed including the MMSE. Patients who met any of the following criteria were not eligible for enrollment: History of seizures, stroke or transient ischemic attack, unstable medical, neurological, or psychiatric condition. Current treatment with typical antipsychotic or neuroleptic medication or within 6 months prior to screening. Anticoagulants within 3 months prior to screening, chronic use of opiates or opioids (including long-acting opioid medication) within 3 months prior to screening, stimulant medications (amphetamine, methylphenidate preparations, or modafinil) within 1 month prior to screening and throughout the study, chronic use of benzodiazepines, barbiturates, or hypnotics. Patients who are legally detained in an official institution or may be dependent on the sponsor, the investigator, or the trial site, individuals without caregiver (lack of a spouse or a close relative that can provide information on the patient's condition), participation in other clinical trials according to AMG (1 month before the time of this trial), and persons showing EEG abnormalities were excluded.

### Randomization and blinding
In this trial, a total of 23 patients were assessed for eligibility (Fig. 4). Twenty patients, who met all inclusion criteria giving written informed consent were enrolled. Randomization was performed uniformly (block randomization, 1:1) and emergency letters were produced by

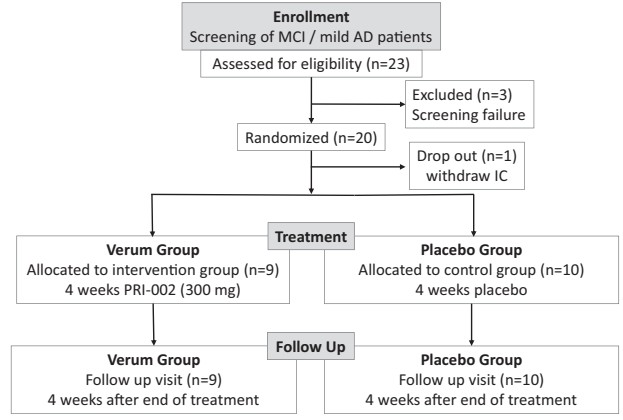

**Fig. 4 | Trial profile.** Patients randomized were prescreened, which resulted in a low number of screening failures. AD: Alzheimer's Disease; IC: Informed consent; MCI: Mild neurocognitive impairment.

the pharmacy of the Charité. All involved parties in this trial (patients and investigators) were blinded to study treatment throughout the whole trial period. Capsules containing PRI-002 and placebo were of identical appearance and were provided by the pharmacy of the Charité, which was also responsible for the random list and emergency envelopes in order to ensure concealment of the blinding procedures.

One patient withdraw informed consent after screening before treatment was initiated. 9 patients were allocated to the verum group and received PRI-002, while consequently 10 patients received placebo treatment. In the study 11 females and 8 males were included, so the findings apply to both sexes. Only sex was considered in study design and sex was determined based on self-reporting. There was no sex specific stratification implemented since no differences concerning efficacy and safety of the investigational product in this regard were expected. The first patient entered the study on 8 December 2020 (first patient in), and the last patient completed the follow-up visit at Day 56 on 13 January 2022 (last patient out). After data cleanup and database lock, the biostatistician was given access to the randomization code and the data were for the first time unblinded.

## Procedures

Patients ($n = 19$) received once daily (q.d.) three capsules each containing 100 mg PRI-002 (in total 300 mg) ($n = 9$) or placebo ($n = 10$), indistinguishable with respect to color, taste, smell, and shape for 28 days. During treatment, the frequency of study visits at the trial site was 14 days (Day 1, day 14, Day 28). A last assessment took place at Day 56. At Day 7 and Day 21 additional phone contacts for safety assessments were performed.

For safety assessment during baseline, at Day 14, 28, and 56, laboratory parameters and vital signs were measured, a physical exam was performed, and ECGs were recorded per protocol. A safety MRI was performed at screening, a second one within 3 days after day 28 and a third one at Day 56. During baseline and at day 28, EEG measurements were performed. In addition, nature, frequency, severity, and timing of AEs and SAEs were recorded at day 14, 28, and 56. Standardized AE reports were provided and analysed by the Data Safety and Monitoring Board (DSMB).

The effect on biomarker was assessed during baseline and after 28 days of medication. CSF collection and blood sampling were performed to determine PRI-002 concentration, which was conducted by Nuvisan (Neu-Ulm, Germany) and to exploratively estimate the effect of PRI-002 on the modulation of Aβ 1-40, Aβ 1-42, Aβ oligomers, total tau (t-tau), and phospho tau (p-tau). Aβ 1-40, Aβ 1-42, t-tau and p-tau$_{181}$ were measured by immunoassay, using the Lumipulse platform (Fujirebio, Tokyo, Japan) and Aβ oligomers concentrations were determined using sFIDA technology[31–33] by attyloid GmbH (Düsseldorf, Germany). The effect of PRI-002 on cognitive and functional performance was measured by repetitively applying the CERAD+ battery and the Clinical Dementia Rating (CDR) at baseline, at Day 28 and additionally at the last visit (Day 56).

The CERAD Word Learning Test is a part of the CERAD cognitive test battery and assesses memory function, particularly for ADe and other dementias[34,35]. It includes three parts: immediate recall, delayed recall, and recognition. During the test administration, a list of 10 words is read aloud three times, with the participant recalling as many as possible after each trial (immediate recall). After a delay of 5–10 min, they attempt to recall the words again (delayed recall), followed by a recognition task where they identify original words from a mixed list (recognition). The scores from the CERAD Word Learning Test are interpreted using normative data, which consider the participant's age, gender, and education level. Normal range scores are considered between the 16th and 84th percentiles (z-score -1 to +1). For MCI scores between the 2nd and 15th percentiles (z-score -1 to -2) are expected.

For significant cognitive impairment scores below the 2nd percentile (z-score < -2) are considered[36].

## Outcomes

Primary objective of the study was the assessment of safety and tolerability of multiple oral doses of PRI-002 in patients with MCI or mild dementia due to AD. Primary endpoints included nature, frequency, severity, and timing of AEs and SAEs; changes in routine laboratory values, ECG, MRI, EEG, and vital signs. Secondary endpoints included the evaluation of pharmacokinetic characteristics of PRI-002 by determination of maximum plasma concentration ($C_{max}$), time to reach maximum plasma concentration ($T_{max}$), half-life ($t_{1/2}$) calculated from PRI-002 plasma concentrations; and the determination of CSF concentrations of PRI-002.

Exploratory objectives included CSF biomarkers for AD pathology in order to assess the effect of PRI-002 on the modulation of Aβ 1-40, Aβ 1-42, Aβ oligomers, t-tau and p-tau. The effect of PRI-002 on cognition and function was tested by the CERAD+ battery and CDR-SB at baseline, Day 28, and Day 56, exploratory endpoints included the change of biomarkers (p-tau, t-tau, Aβ 1-40, Aβ 1-42 and Aβ oligomers) in CSF, blood plasma; change in CERAD+ test battery and CDR-SB scores.

## Statistical analysis

No sample size calculations were done because this trial was exploratory.

All safety parameters and laboratory values (urinalysis, complete blood count (CBC), Quick, partial thromboplastin time (PTT), creatinine, creatinine kinase (CK), creatinine reactive protein (CRP), alanine aminotransaminase (ALT), aspartate aminotransferase (AST)) as well as ECG, EEG, MRI, and vital signs were analyzed in a descriptive manner according to their scales (frequencies, means). In order to compare a variable between both groups (placebo vs PRI-002) at one time point, a parameter free Wilcoxon rank sum test was used. As described in the statistical analysis plan, linear mixed models with random effect were used for the comparison of longitudinal effects between both groups (placebo vs PRI-002). ANOVA was also applied to assess longitudinal effects. Since the results are similar to the linear mixed models, results are presented using linear mixed models. A 95% confidence interval was used to check whether the parameters are within their normal ranges. The $p$ value ($p < 0.05$) for the word list test was not corrected for multiplicity. All statistical data analysis was performed with the certified software Statistical Analysis System (SAS) by MicroDiscovery GmbH (Berlin, Germany).

Cohen's effect size (d) was calculated using the following equations:

$$d = \frac{x_1 - x_2}{pooled\ SD} \tag{1}$$

$$pooled\ SD \sqrt{\frac{(n_1 - 1)s_1^2 + (n_2 - 1)s_2^2}{n_1 + n_2 - 2}} \tag{2}$$

SD: standard deviation
$x_1$: WL learn mean of PRI-002 treated patients
$x_2$: WL learn mean of placebo treated patients
$n_1$: number of PRI-002 treated patients
$s_1$: WL learn SD of PRI-002 treated patients
$n_2$: number of placebo treated patients
$s_2$: WL learn SD of placebo treated patients

## Reporting summary

Further information on research design is available in the Nature Portfolio Reporting Summary linked to this article.

## Data availability

Individual deidentified participant data (including data dictionaries) will be shared; Source data are provided as a source data file or are provided in the supplementary information; additional, related documents (study protocol, statistical analysis plan) will be available upon request to the corresponding author. Source data are provided with this paper.

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

## Acknowledgements

We thank Lutz Frölich (Central Institute of Mental Health, Mannheim, Germany), Holger Stark (Institute of Pharmaceutical and Medicinal Chemistry, Heinrich-Heine-University, Düsseldorf, Germany), Jörg Breitkreutz (Institute of Pharmaceutics and Biopharmaceutics, Heinrich-Heine-University, Düsseldorf, Germany) and Tim Friede (Dept. of Medical Statistics, University Medical Center Göttingen, Göttingen, Germany) for their participation in the study's independent data and safety monitoring board. Statistical Analysis were performed by Chris Bauer (MicroDiscovery GmbH, Berlin, Germany). Dr. Cosma is participant in the BIH Charité Clinician Scientist Program funded the Charité-Universitätsmedizin Berlin, and the Berlin Institute of Health at Charité.

sFIDA experiments were funded by German Federal Agency for Disruptive Innovation (SPRIND). The study was funded by Berlin-Institute of Health (BIH)(O.P.), Forschungszentrum Juelich GmbH (D.W.), SPRIND (O.B., D.W., O.P.). The funding institution had no role in study design, data collection, data analysis, data interpretation, or writing of the report.

## Author contributions

O.P., D.W., J.K. and G.K. contributed to the study concept and design. O.P. (principal investigators at trial sites), N.C.C., C.B. and F.F. contributed to clinical data acquisition. M.P. and O.B. performed and analysed the sFIDA experiments, T.B. performed correlation analysis, O.P., D.W., J.K. and G.K. interpreted the data. O.P., D.W., J.K. and G.K. drafted the manuscript. O.P., D.W., J.K. and G.K. has directly accessed and verified the underlying data reported in the manuscript.

## Funding

## Competing interests

D.W. is cofounder and co-owner of Priavoid GmbH. D.W. is coinventor of PRI-002. O.P. has consulting contracts with Prinnovation GmbH and Priavoid GmbH. This did not have any influence on the interpretation of data. The remaining authors declare no competing interests.

## Additional information

Janine Kutzsche [1], Nicoleta Carmen Cosma[2,3], Gunther Kauselmann [1], Friederike Fenski [2], Christine Bieniek[2], Tuyen Bujnicki[1], Marlene Pils [4], Oliver Bannach[4], Dieter Willbold [1,5] ✉ & Oliver Peters [2] ✉

[1]Institute of Biological Information Processing 7, Structural Biochemistry, Forschungszentrum Jülich GmbH, Jülich, Germany. [2]Department of Psychiatry and Neuroscience, Charité—Universitätsmedizin Berlin, corporate member of Freie Universität Berlin, Humboldt-Universität zu Berlin, Berlin, Germany. [3]Berlin Institute of Health at Charité—Universitätsmedizin Berlin, BIH Biomedical Innovation Academy, BIH Charité Clinician Scientist Program, Berlin, Germany. [4]attyloid GmbH, Düsseldorf, Germany. [5]Institut für Physikalische Biologie, Heinrich-Heine-Universität Düsseldorf, Düsseldorf, Germany. ✉e-mail: d.willbold@fz-juelich.de; oliver.peters@charite.de

