## [Peer Review file · Nature Communications]

Oral PRI-002 treatment in patients with MCI or mild AD: a randomized, double-blinded phase 1b trial.

Corresponding Author: Professor Oliver Peters

Version 0:

Reviewer comments:

Reviewer #1

(Remarks to the Author)

This is report on a phase 1b clinical trial of a novel investigational therapeutic, the anti-oligomeric peptide PRI-002, in CSF biomarker confirmed early AD (MMSE 22-30). The reported major findings were an acceptable safety/tolerability profile and a potential positive efficacy signal on a word-learning test. The compelling aspects of the positive effects were that the results were very consistent, with all nine PRI-002 recipients showing improvement, the magnitude of the effect (3.5 word difference between active drug and placebo) and the inclusion of CSF biomarker positive early AD, patients in whom practice effects are low (see Duff et al, J Prev Alzheimers Dis . 2017;4(2):87-92. doi: 10.14283/jpad.2017.9).

The overarching comment would be that while safety/tolerability is generally (and was in this case) safety and tolerability, that objective should be achieved with a dose that would be reasonably be expected to be clinically active in phase 2 trials. The latter part of that objective is usually achieved through demonstration of target engagement and/or demonstration of a disease-relevant pharmacological effect. Alternatively, the pharmacokinetic data, ideally in CSF, that target drug concentrations (i.e. drug concentrations based on mechanism or preclinical data would be expected to be efficacious) were achieved.

My three major concerns around whether the study objective, stated as in the prior paragraph, were achieved are as follows:

(1) No clear demonstration of target engagement and/pharmacological activity in the brain. Indeed, the protocol did not even have a clearly stated objective in this regard that was specific to the mechanism. In this day and age, that is a flaw in the study design. At the same time, implicit in the mechanism, as well expected based on the preclinical data (where there is an effect on amyloid pathology), is that if target concentrations had been achieved there should have shown some effects on amyloid beta parameters in the CSF. Instead the data mostly argues no effects on a broad set of CSF parameters, that would argue target concentrations were not achieved. The one piece of positive data in this regard is that there is a significant inverse relationship between "slope of oligomeric amyloid beta and plasma drug concentrations. My interpretation of that data is that the dose was too low, as there were only three of nine subjects who showed a reduction (a negative slope) in CSF oligomeric amyloid beta. This potential conclusion must be addressed in the Discussion.

My strong recommendation would that the authors measure baseline and end-of-treatment plasma glial fibrillary acidic protein (GFAP) and plasma ptau217. Both lecanemab and donanemab, both of which target synaptotoxic forms of oligomeric amyloid beta, significantly reduced GFAP and ptau217 in plasma. GFAP is of particular interest as multiple recent reports (e.g. Brain Communications, Volume 6, Issue 4, 2024, fcae218, <https://doi.org/10.1093/braincomms/fcae218>) link astrocyte activation to oligomeric amyloid beta induced synaptic toxicity.

In the absence of target engagement, the Discussion should include a section on the pharmacokinetic result and whether the PK is consistent with having achieved drug concentrations based on the preclinical data. The protocol indicates the expected C_{max} is similar to those achieved in the preclinical studies in which PRI-002 improved cognition. However, most of the time AUC and C_{trough} are more important for efficacy. Are they also comparable? And, in the data in phase 1b (rather than in the MAD)?

(2) The results around the primary objective, safety/tolerability, must be better described. A description of "nervous disorders", seven in active drug and one in placebo is not sufficient; particularly as CNS toxicity was prominent in the nonclinical studies and the CNS was the identified target organ for toxicity. What were these events? Were all the reported related events the same as the nervous system disorders? And, what were the days of onset? Onset late in the 28-day treatment cycle would be problematic as both incidence and severity might increase with continued dosing. Also, depending on the particular event, two events of moderate CNS events that were considered related, would argue this is the maximum tolerated dose, and potentially above MTD, as the severity of events might be expected to be worse in a more advanced, and more heterogeneous, AD patient population.

(3) While there are some reasons to believe the efficacy signal (see my first paragraph), some further information is required. Most importantly, the number of patients in each group who were receiving background therapy should be reported. The difference is between placebo and PRI-002 is as much driven by the three progressors, and there are reports that patients with early AD who are receiving cholinesterase inhibitors show progression, while treatment naive patients show stability. While counter-intuitive, it is thought that the patients who are progressors are identifiable and are placed on cholinesterase inhibitors, while stable patients remain treatment naive. As a result, if there were an imbalance, it might explain the result. It would also be helpful to calculate a Cohen's d effect size (ES), as a ES >0.3 would support a true treatment effect.

Other Comments:

(1) The results of the word list test delayed recall should be reported in the main manuscript and the results of the other components of the CERAD should be provided in a supplement. Delayed recall is most specific to AD, and in particular to oligomeric amyloid beta induced cognitive dysfunction, and needs to be reported along with the total score.

(2) It should be acknowledged that the $p < 0.05$ for the word list test is not corrected for multiplicity. While OK in an exploratory phase 1b study, it needs to be explicitly stated.

(3) The sentence, "Therefore, we planned the word list learning within the CERAD battery of neurocognitive testing for Day 1" should either be modified or deleted. There is nothing in the protocol that indicated that the word list test was specifically selected ahead of any other seven components of CERAD. Rather, the wording in the protocol would indicate the CERAD would be evaluated as a single composite score. The word learning test simply happened to be the one test that showed a significant effect.

(4) Please provide in the methods a description of the CERAD word learning test and how it is administered. And the normal range (and/or cutoff for cognitive impairment), with a reference. It would be helpful to a general reader.

(5) Delete the word "strong" in the sentence "A strong and significant inverse relationship was observed between changes in A β oligomer concentrations..."

(6) Why were there no PK (drug concentration) assessments in CSF? Because BBB penetration in mice vs humans can diverge, matching CSF drug concentrations in human to those achieved in the animal pharmacology studies would be more compelling than matching CSF drug concentrations. If there is a fundamental technical problem, it should be mentioned in the manuscript.

(7) The discussion on adacunumab should be deleted and replaced with a richer discussion around lecanemab, as well on donanemab. Aducanumab is irrelevant, among other reasons it does not have specific activity against synaptotoxic amyloid beta and does not exhibit short term treatment effects. In contrast, lecanemab and donanemab target synaptotoxic amyloid beta (protofibrils and pyroglutamated Ab). Lecanemab also shows initial symptomatic effects, most clearly demonstrated in phase 2b, where there is clear improvement on ADAS-Cog14 at week 12 (Swanson et al, Alzheimer's Research & Therapy volume 13, Article number: 80 (2021))

Reviewer #2

(Remarks to the Author)

This is a well-conducted phase 1 study on Safety, tolerability, and pharmacokinetics of PRI-002 in patients with mild cognitive impairment or mild dementia due to Alzheimer's disease. The study design, analysis methods, and presentation and interpretation of the results are mostly adequate and of a good standard. However, there are still a few issues needing attention.

1. The abstract could be written clearer. For example, it says "During treatment, the frequency of study visits was 14 days. Follow-up assessment was performed on Day 56". There will be 4 visiting points, 14, 28, 42, 56 days. Does it mean the assessment only being performed on Day 56 but not at all on other visit days?

2. The treatment is for 28 days. It is not clear if the AEs and SAEs were counted during the 28-day treatment or during the 56 days follow-up. All these need to be clearly written in the abstract and the main text to avoid any potential ambiguity.

3. There is only one dose level. Have authors considered dose escalation for further groups, which is also typical in phase 1b trials.

Version 1:

Reviewer comments:

Reviewer #1

(Remarks to the Author)

The revision and response largely address the comments by this reviewer. With regard to plasma GFAP, I understand the issues raised by the authors (including acknowledging that 4 weeks may be too short a time period to see an effect), and accordingly withdraw the request as being an absolute necessity. I would though re-state that astrocyte activation, which plasma GFAP is measuring, is intimately linked to the cognitive deficits, and so effects on GFAP may well be seen in parallel with the cognitive effects (improvement on CERAD wordlist) that you are reporting. Also, after reading your response, as well as reading your prior, preclinical publications, it seems that seeing an effect on any of the biomarkers may indeed have been challenging, as in the preclinical studies the most prominent effect is on cognition. The mechanism of action in terms of oligomer/fibril formation is indeed in within the synapse and is likely to be quite dynamic and so it may simply be that this with this mechanism it would not read through in the CSF or plasma. You make such points in the discussion, but its pretty nuanced and you may want to further strengthen this point (for example, I found the title of the 2017 Sci Reo. article compelling: "The A β oligomer eliminating D-enantiomeric peptide RD2 improves cognition without changing plaque pathology").

Then a few editorial comments:

-In figure 4, I am unsure as to what you mean by "slope A β oligomers", is that simply the ratio of Day 28 to Day 1 concentration? If so, better to show data as % change from Day 1 to Day 28. Also, is the "plasma level on Day 1" the Cmax. If so, please label the figure as such. If not, please indicate which plasma level

-on line 414, "has proven that PRI-002 is safe and tolerable" is way too strong after experience with 9 patients for 28 days. Please reword that sentence to read "This phase 1b study met its primary prespecified outcomes with regard to safety and tolerability"

-Line 393, rather than "...provide the first evidence of disease modification", better would be "provide the first evidence of a symptomatic effect that results from disease modification". I believe that was the intent of the publication that is cited. And then delete the sentence that immediately follows regarding shorter latency with PRI-002. We don't have the data to directly compare as in those studies CERAD word list was not utilized in the lecanemab and donanemab studies, and there was not a day 56 assessment of any measure. If there is last sentence to that paragraph it should more along the lines that the collective data argues that short term effects can be achieved by targeting synaptotoxic forms of Abeta, which then I believe makes your data more compelling (i.e more biologically plausible).

Reviewer #2

(Remarks to the Author)

Thanks authors for their effort to improve the manuscript. I am satisfied with the response and revision. No further issues needing attention.

Reviewer: 1

This is report on a phase 1b clinical trial of a novel investigational therapeutic, the anti-oligomeric peptide PRI-002, in CSF biomarker confirmed early AD (MMSE 22-30). The reported major findings were an acceptable safety/tolerability profile and a potential positive efficacy signal on a word-learning test. The compelling aspects of the positive effects were that the results were very consistent, with all nine PRI-002 recipients showing improvement, the magnitude of the effect (3.5 word difference between active drug and placebo) and the inclusion of CSF biomarker positive early AD, patients in whom practice effects are low (see Duff et al, J Prev Alzheimers Dis. 2017;4(2):87-92. doi: 10.14283/jpad.2017.9).

The overarching comment would be that while safety/tolerability is generally (and was in this case) safety and tolerability, that objective should be achieved with a dose that would be reasonably be expected to be clinically active in phase 2 trials. The latter part of that objective is usually achieved through demonstration of target engagement and/or demonstration of a disease-relevant pharmacological effect. Alternatively, the pharmacokinetic data, ideally in CSF, that target drug concentrations (i.e. drug concentrations based on mechanism or preclinical data would be expected to be efficacious) were achieved.

My three major concerns around whether the study objective, stated as in the prior paragraph, were achieved are as follows:

(1) No clear demonstration of target engagement and/pharmacological activity in the brain. Indeed, the protocol did not even have a clearly stated objective in this regard that was specific to the mechanism. In this day and age, that is a flaw in the study design. At the same time, implicit in the mechanism, as well expected based on the preclinical data (where there is an effect on amyloid pathology), is that if target concentrations had been achieved there should have shown some effects on amyloid beta parameters in the CSF.

Instead the data mostly argues no effects on a broad set of CSF parameters, that would argue target concentrations were not achieved. The one piece of positive data in this regard is that there is a significant inverse relationship between "slope of oligomeric amyloid beta and plasma drug concentrations. My interpretation of that data is that the dose was too low, as there were only three of nine subjects who showed a reduction (a negative slope) in CSF oligomeric amyloid beta. This potential conclusion must be addressed in the Discussion.

Answer:

We have added the following text to the discussion (page 14, line 456-463):

"The observation that only three out of nine verum patients showed an absolute reduction of A β oligomer concentration in CSF (negative slope) can potentially be interpreted towards the conclusion that the dose and/or the treatment duration were too low. This will be investigated in the ongoing phase 2 study (<https://clinicaltrials.gov/study/NCT06182085>) with higher dose and longer treatment duration. For the interpretation of any biomarker changes from baseline to day 28, one has to keep in mind that biomarkers usually do not change within 4 weeks, but within months and years, and that biomarkers measured from blood and also from CSF are peripheral to the brain, and it may need more than 28 days to reflect changes in the brain."

(2) My strong recommendation would that the authors measure baseline and end-of-treatment plasma glial fibrillary acidic protein (GFAP) and plasma ptau217. Both lecanemab and donanemab, both of which target synaptotoxic forms of oligomeric amyloid beta, significantly reduced GFAP and ptau217 in plasma. GFAP is of particular interest as multiple recent reports (e.g. Brain Communications, Volume 6, Issue 4, 2024, fcae218, <https://doi.org/10.1093/braincomms/fcae218>) link astrocyte activation to oligomeric amyloid beta induced synaptic toxicity.

Answer: As already stated in the answer to question #1 AD biomarkers usually do not change within 4 weeks, but within several months or even years, and that biomarkers measured from blood and also from CSF are peripheral to the brain and it may need more than 28 days to reflect changes in the brain. This is most likely the reason why the first measurements for lecanemab were performed after 6 months of treatment and for donanemab after 12 weeks of treatment (see figures below). Therefore, we do not expect any significant changes in the very peripheral blood within 28 days for plasma GFAP or plasma ptau₂₁₇. We still have one last plasma sample from each patient left that would allow the measurement of both markers, but this would mean that no further material would be available for future questions. Of course, the reviewer could not know this dilemma. Therefore, we ask to withdraw this request. Should the reviewer nevertheless insist, we are prepared to carry out the measurements.

[REDACTED]

<https://www.alzforum.org/news/conference-coverage/dare-we-say-consensus-achieved-lecanemab-slows-disease>

[REDACTED]

[1] and <https://www.alzforum.org/news/research-news/trailblazer-plasma-gfap-falls-nfl-continues-rise>

(3) In the absence of target engagement, the Discussion should include a section on the pharmacokinetic result and whether the PK is consistent with having achieved drug concentrations based on the preclinical data. The protocol indicates the expected C_{max} is similar to those achieved in the preclinical studies in which PRI-002 improved cognition. However, most of the time AUC and C_{trough} are more important for efficacy. Are they also comparable? And, in the data in phase 1b (rather than in the MAD)?

Answer: We agree with the reviewer that the AUC would have been a more important measure of PK than only the C_{max}. Unfortunately, it was not possible to calculate this value as it requires multiple blood sampling timepoints. For animal welfare reasons, only one blood sample was taken from a mouse at a time. To add three more mice per sampling timepoint to the treatment study (30 tg animals in total) was not possible, as the number of tg animals is always very limited. To determine the C_{S_{through}}, another blood sample (predose) would also have been required, which was not possible again due to animal welfare. Therefore, we can not say, if these values (AUC and C_{S_{through}}) are comparable between the animal data (efficacy studies with tg mice) and the phase 1b human data. Although the C_{max} is possibly not the most meaningful measure, we could show as the reviewer stated, that the mean C_{max}-value in the MAD study in healthy volunteers at the highest dose was in the concentration range that resulted in significant improvement in cognitive performance in the highest dosed animal disease model. As requested we included the following section to the discussion page 14 line 463-465: "The mean C_{max}-value at day 28 (12.6 ± 13.2 ng/ml) was in the concentration range that resulted in significant improvement in cognitive performance of 200 mg/kg treated old aged APP/PS1delta E9 mice (range of 1.5 ng/ml ± 7.5 ng/ml)[2]."

(4) The results around the primary objective, safety/tolerability, must be better described. A description of "nervous disorders", seven in active drug and one in placebo is not sufficient; particularly as CNS toxicity was prominent in the nonclinical studies and the CNS was the identified target organ for toxicity. What were these events? Were all the reported related events the same as the nervous system disorders? And, what were the days of onset? Onset late in the 28-day treatment cycle would be problematic as both incidence and severity might increase with continued dosing. Also, depending on the particular event, two events of moderate CNS events that were considered related, would argue this is the maximum tolerated dose, and potentially above MTD, as the severity of events might be expected to be worse in a more advanced, and more heterogenous, AD patient population.

Answer: We included the following sentence in the results part on page 9 line 284-289: "One patient in the verum group reported 6 times short-term dizziness on Day 4, Day 8, Day 9, Day 10, Day 18 and Day 20. Another patient in the verum group reported once slight fatigue combined with short-term dizziness on Day 13 and twice fatigue simultaneous to administration of Donepezil on Day 20 and Day 21. Altogether the described events were rated as probably related but not considered as critical in any regard. Especially there is no evidence of an increase of number or severity of adverse events over time.

(5) While there are some reasons to believe the efficacy signal (see my first paragraph), some further information is required. Most importantly, the number of patients in each group who were receiving background therapy should be reported. The difference is between placebo

and PRI-002 is as much driven by the three progressors, and there are reports that patients with early AD who are receiving cholinesterase inhibitors show progression, while treatment naive patients show stability. While counter-intuitive, it is thought that the patients who are progressors are identifiable and are placed on cholinesterase inhibitors, while stable patient remain treatment naive. As a result, if there were an imbalance, it might explain the result. It would also be helpful to calculate a Cohen's d effect size (ES), as a ES >0.3 would support a true treatment effect.

Answer: The reviewer is right; this important information was missing. We included an additional table into the supplement and the following sentence in the results part page 11 line 345-347: "In addition, we analyzed a possible relationship between the treatment with cholinesterase number of patients and duration of treatment on one hand and the improvement of memory function as measured by the CERAD word list (supplement Figure S6)." 6 patients of the PRI-002 group and 8 patients of the placebo group received background therapy. So, less patients with a longer total treatment period received PRI-002, therefore a bias as described by the reviewer appears to be unlikely.

In addition, we calculated the Cohen's d effect size ES:0.94 and included the following sentence in the results part page 11 line 336-337: "The Cohen's d effect size for day 56 was ES:0.94, indicating a large effect and therefore supporting a true treatment effect.". We also added the calculation of the ES to the material and methods part page:6, line 200-211.

Cohen's effect size (ES) was calculated using the following equations:

$$ES = \frac{x_1 - x_2}{\text{pooled } SD}$$

$$\text{pooled } SD = \sqrt{\frac{(n_1 - 1)s_1^2 + (n_2 - 1)s_2^2}{n_1 + n_2 - 2}}$$

SD: standard deviation

x₁: WL learn mean of PRI-002 treated patients

x₂: WL learn mean of placebo treated patients

n₁: number of PRI-002 treated patients

s₁: WL learn SD of PRI-002 treated patients

n₂: number of Placebo treated patients

s₂: WL learn SD of Placebo treated patients

Patient	Treatment	Difference WL learn Baseline (day 1) to EoS (day 56)	Duration cholinesterase inhibitor treatment [month]	Diagnosis
1	PRI-002	7	3	MCI
2	PRI-002	4		MCI
3	PRI-002	2	10	MCI
4	PRI-002	5		MCI
5	PRI-002	1		MCI
6	PRI-002	4	3	MCI
7	PRI-002	6	30	AD
8	PRI-002	3	3	MCI

9	PRI-002	3	4	AD
			Σ 53	
1	Placebo	-3	7	MCI
2	Placebo	0	3	MCI
3	Placebo	5		MCI
4	Placebo	0	6	MCI
5	Placebo	6		MCI
6	Placebo	-2	3	MCI
7	Placebo	2	3	MCI
8	Placebo	5	3	MCI
9	Placebo	-2	3	MCI
10	Placebo	1	3	MCI
			Σ 31	
Table S6: Difference WL learn, Duration cholinesterase inhibitor treatment and Diagnosis; Σ : Total months of treatment				

(6) The results of the word list test delayed recall should be reported in the main manuscript and the results of the other components of the CERAD should be provided in a supplement. Delayed recall is most specific to AD, and in particular to oligomeric amyloid beta induced cognitive dysfunction, and needs to be reported along with the total score.

Answer: The results of the word list test delayed recall were included in Table 4 of the main manuscript and an additional Table S6 was included in the supplement to provide all components of the CERAD battery.

Variable	Visit	Placebo			PRI-002			PV	PV-LME
		Mean	SD	n	Mean	SD	n		
WL learn	Day 1	14.30	4	10	15.44	5	9	0.743	0.036
	Day 28	15.33	4	9	17.25	5	8	0.923	
	Day 56	15.50	3	10	19.33	5	9	0.036	
WL Recall	Day 1	2.4	1.7	10	4.7	2.3	9	0.04	0.17
	Day 28	3.2	2.2	9	6.0	3.3	8	0.07	
	Day 56	4.2	1.4	10	5.4	2.2	9	0.39	
CDR-SB	Day 1	1.56	1.01	9	2.93	2.26	7	0.26	0.484
	Day 28	1.31	0.46	8	1.75	1.89	8	0.75	
	Day 56	1.79	0.70	7	2.39	2.23	9	0.96	
MMSE	Day 1	26.9	1.91	10	27.1	2.93	9	0.535	0.730
	Day 28	26.4	2.40	9	27.9	2.36	8	0.261	
	Day 56	27.3	1.83	10	27.2	2.39	9	0.900	

P-values (PV) for one time point (column 'PV') were calculated using Wilcoxon test. PV for the comparison of longitudinal effects between both groups (Placebo vs PRI-002) (column 'PV-LME') where calculated with mixed linear models. Day 1 is the baseline visit, Day 28 is the end of treatment visit and Day 56 is the End of Study visit.

n: Number; SD: Standard deviation; WL learn: CERAD Word list learning; WL Recall: Word list recall test; CDR-SB: Clinical Dementia Rating scale Sum of Boxes I; MMSE: Mini mental state examination.

Table 4: Statistics for functional outcome measures

Variable	Day	Placebo			PRI-002			PV	PV-LME
		mean	SD	N	mean	SD	N		
TMT-A	1	45.3	13.5	10	60.4	26.7	9	0.24	0.49
	28	45.8	11.4	9	58.4	23.9	8	0.31	
	56	43.7	9.9	10	53.9	21.1	9	0.46	
TMT-B	1	121.5	43.7	10	157.6	72.1	9	0.27	0.23
	28	113.3	25.3	9	137.5	70.2	8	0.74	
	56	112.3	34.0	10	121.9	56.1	9	0.87	
Fig Draw	1	10.6	0.7	10	10.3	1.1	9	0.77	0.09
	28	10.9	0.3	9	10.8	0.7	8	0.93	
	56	10.6	1.0	10	10.9	0.3	9	0.61	
Fig Recall	1	4.2	3.2	10	5.7	3.4	9	0.37	0.64
	28	5.6	3.0	9	6.9	3.8	8	0.41	
	56	6.6	3.0	10	7.4	4.4	9	0.43	
Clock	1	1.6	0.8	10	1.7	0.9	9	0.89	0.78
	28	1.6	0.9	9	1.6	1.1	8	0.95	
	56	1.7	0.8	10	1.7	1.1	9	0.71	
WL learn	1	14.3	3.7	10	15.4	4.6	9	0.74	0.04
	28	15.3	4.3	9	17.3	5.0	8	0.92	
	56	15.5	3.4	10	19.3	4.7	9	0.04	
WL Recall	1	2.4	1.7	10	4.7	2.3	9	0.04	0.17
	28	3.2	2.2	9	6.0	3.3	8	0.07	
	56	4.2	1.4	10	5.4	2.2	9	0.39	

WMS delayed recall	1	2.1	2.3	10	10.0	11.6	9	0.09	0.58
	28	3.3	6.0	9	12.8	12.0	8	0.05	
	56	5.8	6.1	10	15.0	13.6	9	0.14	
P-values for one time point (column 'PV') were calculated using wilcoxon test. P-values for the comparison of longitudinal effects between both groups (Placebo vs. PRI-002) (column 'PVLME') where calculated with mixed linear models. Day1 is the baseline visit, day 28 is the end of treatment visit and day 56 is the end of study visit. TMT-A: Tests Trail Making Test A, TMT-B: Tests Trail Making Test B, Fig Draw: Figure Drawing test, Fig Recall: Figure Recall test, Clock: Clock drawing test, WL learn: word list learning test, WL Recall: Word list recall test, WMS: Wechsler Memory Scale delayed recall test.									
Table S6 with statistics for CERAD variables.									

(7) *It should be acknowledged that the $p < 0.05$ for the word list test is not corrected for multiplicity. While OK in an exploratory phase 1b study, it needs to explicitly stated.*

Answer: We included the sentence to the methods part page 6 line 197-198: “The p-value ($p < 0.05$) for the word list test was not corrected for multiplicity.”

(8) *The sentence, "Therefore, we planned the word list learning within the CERAD battery of neurocognitive testing for Day1" should either be modified or deleted. There is nothing in the protocol that indicated that the word list test was specifically selected ahead of any other seven components of CERAD. Rather, the wording in the protocol would indicate the CERAD would be evaluated as a single composite score. The word learning test simply happened to be the one test that showed a significant effect.*

Answer: We deleted the sentence as requested.

(9) *Please provide in the methods a description of the CERAD word learning test and how it is administered. And the normal range (and/or cutoff for cognitive impairment), with a reference. It would be helpful to a general reader.*

Answer: We included the following part in the methods section page 5, line 160-170. “The (Consortium to Establish a Registry for Alzheimer's Disease) CERAD Word Learning Test is a part of the CERAD cognitive test battery and assesses memory function, particularly for Alzheimer's disease and other dementias [3, 4]. It includes three parts: immediate recall, delayed recall, and recognition. During the test administration, a list of 10 words is read aloud three times, with the participant recalling as many as possible after each trial (immediate recall). After a delay of 5–10 minutes, they attempt to recall the words again (delayed recall), followed by a recognition task where they identify original words from a mixed list (recognition). The scores from the CERAD Word Learning Test are interpreted using normative data, which consider the participant's age, gender, and education level. Normal range scores are considered between the 16th and 84th percentiles (z-score -1 to +1). For mild cognitive impairment scores between the 2nd and 15th percentiles (z-score -1 to -2) are expected. For significant cognitive impairment scores below the 2nd percentile (z-score < -2) are considered [5].”

(10) Delete the word "strong" in the sentence "A strong and significant inverse relationship was observed between changes in A β oligomer concentrations..."

Answer: done

(11) Why were there no PK (drug concentration) assessments in CSF? Because BBB penetration in mice vs humans can diverge, matching CSF drug concentrations in human to those achieved in the animal pharmacology studies would be more compelling than matching CSF drug concentrations. If there is a fundamental technical problem, it should be mentioned in the manuscript.

Answer: We did not report the CSF PRI-002 concentrations, because of a technical problem, as we were unable to determine exact values. Due to non-specific binding of PRI-002 to vials used for sampling and storage of CSF, the measured values, which are depicted in the Table below are minimal values and will probably be higher than reported here.

Considering the three CSF samples in which the PRI-002 levels were above the LLOQ, approximately 15% of the mean predose plasma level of all patients at Day 28 (1.97 ± 2.4 ng/ml) reached the CSF and 8% of the mean predose plasma level of patient (2-4) at Day 28 (3.74 ± 3.42 ng/ml) reached the CSF.

We will include the Table S5 in the supplement and the following sentence in the results part page 11, line 327-331: "In addition, also CSF levels of PRI-002 were determined. Due to non-specific binding of PRI-002 to vials used for sampling and storage the measured values, which are depicted in the Table S7 are minimal values and the actual values will most probably be higher than reported here. Considering the three CSF samples in which the PRI-002 levels were above the LLOQ, approximately 8% of the mean predose plasma level of patients 2, 3 and 4 at Day 28 (3.74 ± 3.42 ng/ml) reached the CSF."

Patient PRI-002	ng PRI-002/ml
1	<0.200
2	0.203
3	0.377
4	0.304
5	<0.200
6	<0.200
7	<0.200
8	<0.200
9	<0.200
LLOQ:0.200; <0.200: below LLOQ	
Table S7 PRI-002 Level in CSF of individual patients at Day 28	

(12) The discussion on aducunumab should be deleted and replaced with a richer discussion around lecanamab, as well on donanemab. Aducanumab is irrelevant, among other reasons it does not have specific activity against synaptotoxic amyloid beta and does not exhibit short term treatment effects. In contrast, lecanemab and donanemab target synpatotoxic amyloid beta (protofibrils and pyroglutamated Ab). Lecanamab also shows initial symptomatic

effects, most clearly demonstrated in phase 2b, where there is clear improvement on ADAS-Cog14 at week 12 (Swanson et al, Alzheimer's Research & Therapy volume 13, Article number: 80 (2021)

Answer: We agree with the reviewer regarding the in part identical targets of lecanemab and PRI-002 and therefore emphasize the similarities they might have. We thank the reviewer for his recommendation to cite Swanson et al. 2021 and highlight now the symptomatic improvement as measured by ADAS-cog. Aducanumab partially targets A-beta oligomers (Tolar et al., Alzheimers Res Ther, 2020; Pernecky et al., Brian, 2023) and might therefore also have at least some specific activity against synaptotoxic amyloid species. We have reduced the weighting of aducanumab and added donanemab in the discussion page 13 line 396-401: "Interestingly Lecanemab 10 mg biweekly has also shown a symptomatic improvement as measured by ADAS-cog from week 12[6] and most prominent at 6 month after treatment was initiated[7]. Donanemab (700 mg for the first 3 doses and 1400 mg thereafter administered intravenously every 4 weeks) even demonstrated a significant improvement after 12 weeks of treatment in the ADAS-Cog₁₃ and the CDR-SB. These clinical efficacy signals, demonstrated by improvement in symptoms combined with significant changes in biomarkers, provide the first evidence of disease modification[6]."

Reviewer: 2

This is a well-conducted phase 1 study on Safety, tolerability, and pharmacokinetics of PRI-002 in patients with mild cognitive impairment or mild dementia due to Alzheimer's disease. The study design, analysis methods, and presentation and interpretation of the results are mostly adequate and of a good standard. However, there are still a few issues needing attention.

1. The abstract could be written clearer. For example, it says "During treatment, the frequency of study visits was 14 days. Follow-up assessment was performed on Day 56". There will be 4 visiting points, 14, 28, 42, 56 days. Does it mean the assessment only being performed on Day 56 but not at all on other visit days?

Answer: We thank the reviewer for his comment and have checked the abstract for inaccurate wording. We have rephrased the sentences on page 1 line 27-30: "During treatment, study visits were performed on baseline (Day 1), Day 14, Day 28 and an additional follow-up visit on Day 56. Safety assessments were carried out at all visits to determine the primary endpoints. On Day 7 and Day 21 additional phone visit were carried out to assess concomitant meds and AEs."

2. The treatment is for 28 days. It is not clear if the AEs and SAEs were counted during the 28-day treatment or during the 56 days follow-up. All these need to be clearly written in the abstract and the main text to avoid any potential ambiguity.

Answer: The safety assessment was carried out at all visits and at the follow-up visit to determine the primary endpoints, so the AEs and SAEs were counted during the entire study duration (day 1 - day 56). See rephrased sentence question #1.

3. There is only one dose level. Have authors considered dose escalation for further groups, which is also typical in phase 1b trials

Answer: We agree, that a dose escalation would have been interesting and helpful. Because, we intended to demonstrate safety in the first place, we applied the highest dose that was used in the previous phase 1 study with healthy volunteers[8] . A dose escalation is done in the ongoing phase 2 study (<https://clinicaltrials.gov/study/NCT06182085>) with higher dose and longer treatment duration.

1. Pontecorvo, M.J., et al., *Association of Donanemab Treatment With Exploratory Plasma Biomarkers in Early Symptomatic Alzheimer Disease: A Secondary Analysis of the TRAILBLAZER-ALZ Randomized Clinical Trial*. *JAMA Neurology*, 2022. **79**(12): p. 1250-1259.
2. Schemmert, S., et al., *Abeta Oligomer Elimination Restores Cognition in Transgenic Alzheimer's Mice with Full-blown Pathology*. *Mol Neurobiol*, 2018.
3. Welsh, K.A., et al., *The Consortium to Establish a Registry for Alzheimer's Disease (CERAD). Part V. A normative study of the neuropsychological battery*. *Neurology*, 1994. **44**(4): p. 609-14.
4. Chandler, M.J., et al., *A total score for the CERAD neuropsychological battery*. *Neurology*, 2005. **65**(1): p. 102-6.
5. Luck, T., et al., *Age-, sex-, and education-specific norms for an extended CERAD Neuropsychological Assessment Battery-Results from the population-based LIFE-Adult-Study*. *Neuropsychology*, 2018. **32**(4): p. 461-475.
6. Alam, J. and M.N. Sabbagh, *Perspective: Minimally clinically important "symptomatic" benefit associated with disease modification resulting from anti-amyloid immunotherapy*. *Alzheimers Dement (N Y)*, 2025. **11**(1): p. e70035.
7. Swanson, C.J., et al., *A randomized, double-blind, phase 2b proof-of-concept clinical trial in early Alzheimer's disease with lecanemab, an anti-A β protofibril antibody*. *Alzheimers Res Ther*, 2021. **13**(1): p. 80.
8. Kutzsche, J., et al., *Safety and pharmacokinetics of the orally available antiprionic compound PRI-002: A single and multiple ascending dose phase I study*. *Alzheimers Dement (N Y)*, 2020. **6**(1): p. e12001.

Response to Reviewer

Reviewer: 1

The revision and response largely address the comments by this reviewer. With regard to plasma GFAP, I understand the issues raised by the authors (including acknowledging that 4 weeks may be too short a time period to see an effect), and accordingly withdraw the request as being an absolute necessity. I would though re-state that astrocyte activation, which plasma GFAP is measuring, is intimately linked to the cognitive deficits, and so effects on GFAP may well be seen in parallel with the cognitive effects (improvement on CERAD wordlist) that you are reporting. Also, after reading your response, as well as reading your prior, preclinical publications, it seems that seeing an effect on any of the biomarkers may indeed have been challenging, as in the preclinical studies the most prominent effect is on cognition. The mechanism of action in terms of oligomer/fibril formation is indeed in within the synapse and is likely to be quite dynamic and so it may simply be that this with this mechanism it would not read through in the CSF or plasma. You make such points in the discussion, but its pretty nuanced and you may want to further strengthen this point (for example, I found the title of the 2017 Sci Reo. article compelling: "The A β oligomer eliminating D-enantiomeric peptide RD2 improves cognition without changing plaque pathology".

1. In figure 4, I am unsure as to what you mean by "slope A β oligomers", is that simply the ratio of Day 28 to Day 1 concentration? If so, better to show data as % change from Day 1 to Day 28. Also, is the "plasma level on Day 1" the C_{max}. If so, please label the figure as such. If not, please indicate which plasma level

Answer: We included a definition of slope A β oligomers in the figure legend: "The slope corresponds to a difference normalized by visit ($0.5 * \Delta \text{Day56} - \text{Day 1}$) and changed the label to C_{max} PRI-002 plasma level day 1 [ng/ml].

2. on line 414, "has proven that PRI-002 is safe and tolerable" is way too strong after experience with 9 patients for 28 days. Please reword that sentence to read "This phase 1b study met its primary prespecified outcomes with regard to safety and tolerability"

Answer: We reword that sentence to "This phase 1b study met its primary prespecified outcomes with regard to safety and tolerability".

2. Line 393, rather than "...provide the first evidence of disease modification", better would be "provide the first evidence of a symptomatic effect that results from disease modification".

Answer: We rephrased that sentence to "provide the first evidence of a symptomatic effect that results from disease modification".

4. I believe that was the intent of the publication that is cited. And then delete the sentence that immediately follows regarding shorter latency with PRI-002.

Answer: We deleted the sentence “Therefore, if we compare the latency of clinical efficacy between anti-A β -antibodies and PRI-002 we speculate that the shorter latency of clinical improvement in our phase 1b study maybe due to the more effective neutralization of synaptic toxic A β oligomers.” as requested.

5. We don't have the data to directly compare as in those studies CERAD word list was not utilized in the lecanemab and donanemab studies, and there was not a day 56 assessment of any measure. If there is last sentence to that paragraph it should more along the lines that the collective data argues that short term effects can be achieved by targeting synaptotoxic forms of Abeta, which then I believe makes your data more compelling (i.e more biologically plausible).

Answer: We included the following sentence in the discussion page 8 line 404-405: “The collective data suggest that short-term effects can be achieved by targeting synaptotoxic forms of Abeta.”